# Accurate Preoperative Localization of Thoracolumbar Disc Extrusion in Dogs: A Prospective Controlled Study

**DOI:** 10.3390/vetsci11090434

**Published:** 2024-09-13

**Authors:** William McCartney, Ciprian Ober, Christos Yiapanis

**Affiliations:** 1North Dublin Orthopaedic Animal Hospital, D13 K5H0 Dublin, Ireland; billymccartney@gmail.com; 2Faculty of Veterinary Medicine, Department of Surgery, University of Agricultural Sciences and Veterinary Medicine, 400372 Cluj-Napoca, Romania; 3Cyvets Veterinary Centre, Paphos 8025, Cyprus; drcy@cyvets.com

**Keywords:** disc disease, error, dogs, thoracolumbar

## Abstract

**Simple Summary:**

Two of the most common errors in spine surgery in humans are performing wrong-level surgery (WLS) and/or performing wrong-sided surgery (WSS). In the present study, we identified the factors contributing to WLS and WSS in a population of eighty-five dogs. Whether the surgeon was able to preoperatively pinpoint the correct site for the surgery was assessed using two small Backhaus clamps moved until correct placement was radiologically confirmed. Utilizing this technique avoided operating on the wrong spine dog and performing the wrong procedure. To improve the accuracy of the incision site choice, the surgeon might use the method presented in this study to determine the exact lesion location prior to surgery. Careful recognition of the potential pitfalls contributing to WLS or WSS as presented in this study should better warn surgeons to limit such errors in spinal surgeries in dogs.

**Abstract:**

Intervertebral disc disease (IVDD) is a very common cause of spinal cord compression in dogs. The errors of human surgeons operating on the wrong level or wrong side is a recognized problem and has been largely analyzed. To date, little information is available regarding these errors in dogs. The objective of this study was to assess the accuracy of disc localization prior to possible surgery for IVDD to ensure the surgeon begins their incision directly over the extruded disc. Eighty-five chondrodystrophic or brachycephalic dogs that presented with IVDD confirmed by CT or MRI scan were included in the study. Two small Backhaus clamps were placed cranial and caudal to the lesion site and a control clamp placed at the wing of the ilium. The main interest was whether the surgeon was able to preoperatively pinpoint the correct site for the surgery. Dorsoventral radiographs were taken to verify by another person if the clamps had been placed in the correct position. If the result was incorrect, the surgeon was asked to try again without knowing that the clamps were incorrect, and another radiograph was taken. This was repeated until the position was correct. The results were recorded as correct or incorrect and the number of attempts were registered. The results suggested no significant trend over time for any of the outcomes examined. There were some slight improvements over time, but none of the results was close to statistical significance. The findings of the study showed that in the thoracolumbar region the surgeon has a higher chance of incorrectly marking the exact site for surgery.

## 1. Introduction

In human spinal surgery, the errors of surgeons operating on the wrong level (WLS) or wrong side (WSS) is a recognized problem and has been discussed in peer review publications [1,2,3,4,5,6]. These errors are considered a unique pitfall of spinal surgery.

The errors in human spinal surgery are caused by human error or anatomical variation amongst other factors. Fifty percent of spinal surgeons admitted operating on the wrong site at least once through their career [4]. Not following verification protocols is a common factor. Additional protocols have been introduced to specifically prevent the occurrence of WLS or WSS, and they have reduced it, but it is still a problem [5]. Once an error has been made, it can lead to permanent problems or litigation.

Wrong-level or wrong-side surgery reports are rarely mentioned in the veterinary literature, likely due to biased reporting. There are only two reports where this is mentioned [7,8]. One report [7] highlighted the consequence of a bilateral mini-hemilaminectomy and intervertebral disc fenestration at T12-13 together with a bilateral pediculectomy of T13. Secondary subluxation and spinal cord compression five days postoperatively required stabilization with bilateral articular screws and a dorsal spinal plate. The conclusion of the second report [8] was that operating on the wrong side or level does not always require a second revision surgery from the correct side to achieve spinal decompression.

Thus, the objective of this study was to analyze how the surgeon decided on the correct approach to the site for surgery in confirmed spinal cases.

## 2. Materials and Methods

### 2.1. Population of the Study

Breed, age, sex, neurological examination findings and duration of clinical signs prior to presentation were registered (Table 1). Dogs with complete medical records, neurological examination and spinal cord compression diagnosed by CT (n = 60) (Figure 1) or MRI (n = 25) during year 2022 and 2023 were included in this study (Table 1). Specific neurolocalization (intervertebral space number) was represented by T10-T11 (n = 5), T11-T12 (n = 31), T12-T13 (n = 23), T13-L1 (n = 16), L1-L2 (n = 5), L2-L3 (n = 3) and L3-L4 (n = 2). A board-certified surgeon (WMC) and two residency-trained surgeons (CO and CY) performed neurological localization attempts and surgeries.

All necessary institutional guidelines for the use of animals were followed. Informed consent of the owners was obtained prior to neurological examination, imaging studies and surgery.

### 2.2. Study Protocol

With the dogs still anesthetized after the CT or MRI scan and the site of the significant disc extrusion confirmed (e.g., T13-L1 left caudal aspect of T13 vertebral body), the surgeon was asked to confirm the site for surgery by placing 2 small Backhaus clamps cranial and caudal to the lesion site using palpation of bone landmarks (e.g., last rib, dorsal spinal process). Any lesion from T8 to L1 was deemed to be thoracolumbar and lumbar if between L1-L7. Backhaus clamps were used as they would not move, unlike needles. The clamps would be close together. As a control, a clamp was also placed at the wing of the ilium. The dog was then x-rayed in a dorsoventral position to verify if the clamps had been placed in the correct position by another person. If the result was incorrect, the surgeon was asked to try again without knowing how the clamps were incorrect, and another radiograph was taken. This was repeated until the position was correct. The results were recorded as correct or incorrect and number of attempts were noted. A correct result meant that a line drawn directly from the centre between the 2 clamps perpendicular to the spine was exactly at the disc lesion.

### 2.3. Statistical Analysis

Subjects recruited into the study were split into groups based on the order in which they underwent surgery. The analyses examined if there was a trend towards different outcomes at further subjects underwent surgery. The first outcomes were the number of subjects with a correct identification of site after 1, 2, 3 and 4 attempts. This was measured cumulatively, so that a subject with a correct identification after one attempt was by default correct after two, three and four attempts. Analyses were performed to compare the characteristics of the different breeds. There were only a small number of dogs of some breeds, and thus to increase the numbers in each category those breeds with small numbers were combined together for analysis. Comparisons of age group and gender were compared between breeds using the Chi-square test. Grade was considered to be an ordinal variable, and to take account of the order of the categories, the analysis was performed using the Kruskal–Wallis test.

## 3. Results

### 3.1. Thoracolumbar Data

The first analyses considered the thoracolumbar data. The analyses were performed to summarize the study data and examine whether there was a trend across the course of the study. The analysis results are summarized in Table 2. The number and percentage of correct site identifications after 1, 2, 3 and 4 attempts are highlighted. The mean and standard deviation number of attempts are presented to obtain a correct siting within each group. The *p*-Values from the analyses, indicates the significance of the trend across the four sequential groups.

The results indicated statistically significant trend for the percentage of subjects with a correct identification of surgical site after one, two and three attempts. No formal comparison was performed for attempt 4, as all sites were correctly identified in all four groups. After one attempt, only 27% of subjects in the first set of cases had a correctly identified site. The equivalent figure increased to 64% in the final set of cases. After two attempts, only 47% of the first set of cases had a correctly identified site, rising to 86% of subjects in the final set of cases. The number of attempts to obtain a correct site also varied significantly over the four groups of cases. The mean number of attempts fell from 2.5 for the first set of cases, down to 1.5 after the final set.

### 3.2. Lumbar Data

A similar set of analyses were performed for the lumbar surgeries. A summary of the analysis results is given in Table 3.

## 4. Discussion

Wrong-site surgery and wrong-level surgery are reported in human neurosurgery, but currently veterinary neurosurgery has no specific reports related to incidence and management of such situations even if disc hernia is one of the most common neurologic problems encountered in veterinary clinical practice. [9]. Thus, the objective of this study was to assess the accuracy of disc localization prior to possible surgery for disc disease to ensure the surgeon begins their incision directly over the extruded disc. Of particular interest was whether surgeons would be able to preoperatively pinpoint the correct site for the surgery.

The results of our study suggested there is no significant trend over time for any of the outcomes examined. There were some slight improvements over time, but none of the results was close to statistical significance. The surgeons were able to have sequential multiple attempts until they obtained the site correct. Fewer errors in the assessment of neurolocalization occurred in the distal intervertebral spaces of the thoracic spine.

A surgeon relies on the diagnostic imaging to verify where spine requires decompression. Presuming this imaging is the correctly labelled then the site is precisely determined if using cross-sectional imaging. Next, the patient must be prepared for surgery and then the surgeon must incise the skin and approach the spine at the precise site determined by cross-sectional imaging. It is at the stage when the surgeon makes the incision that the error of WLS or WSS is made. To determine where to make the incision, the surgeon uses palpable landmarks such as dorsal spinal process and ribs. This is somewhat prone to error because of anatomical variation and in obese patients [5,10]. But even without either of the above, the use of external landmarks is not as accurate as one would presume based on the results of this study. Now add in human error and the chances of a WLS or WSS are greater. Furthermore, there are no established protocols for WLS or WSS avoidance verification in veterinary surgery.

Related to demographic characteristics of the dogs in the study, and comparisons between the different breeds, the analysis results indicated no significance between the different breeds for either age, gender or grade. WLS would, in the authors opinion, be a common event. If a WLS has been made, then this can be easily corrected intraoperatively by extending the incision. This would entail extending the completed ostectomy though further hemilaminectomy. Further surgical trauma occurs and the operating time increases. Increasing the surgical trauma will lead to greater postoperative morbidity and a slower recovery. The recovery from spinal surgery is already known for its extended time, so extending that period will impair recovery. Minimally invasive surgery preserves the soft tissue envelope and diminishes inflammation and dysfunction [11]. Therefore, the aim of the surgery should be, in the scenario of removing a disc extrusion from a specific location (e.g., T13-L1), to create the hemilaminectomy window directly over the extruded disc with minimal bone removal. At times, if this is achieved, then the disc material will, if under pressure, seep out of the bone incision. Thus, the prevention of WLS is a surgical goal, and this can be best achieved by using the technique described in this paper.

WSS is a different matter and involves incising the wrong side completely. Although not catastrophic and the same incision can be still used [8], the route to complete the surgery is not ideal or preferred. WSS can still achieve spinal cord decompression but again, not unlike WLS, it involves more surgical trauma and the knock-on effects to recovery.

To prevent WLS again, the technique described in this paper provides a method to avoid this occurrence. In human medicine, there are a few recommendations to prevent WLS, such as communication with the patient, marking of the intended site and the use of intraoperative radiograph. [4]. This study showed that the technique for marking the intended site with novel solution is a viable source of prevent errors. Intraoperative radiographs is another potential solution for veterinary patients and future studies need to validate it.

There could be a couple of reasons for the lack of statistical significance for lumbar data. Firstly, lumbar surgeries had better results after one attempt for the first set of cases when compared to the thoracolumbar data, giving less room for improvement. Also, the number of subjects in each group was smaller, and thus there was relatively lower statistical power to show a difference.

## 5. Conclusions

In the thoracolumbar region, the surgeon has a higher chance of incorrectly marking the exact site for surgery. To minimize the incision size and therefore surgical trauma, the exact site for incision should be chosen over the extruded disc. Using standard landmarks is not a completely reliable method in the thoracolumbar region for determining the site for incision. The problem of incision site accuracy is exacerbated by being overweight, unusual anatomical variations and in cases with deformities. To improve the accuracy of the incision site choice, the surgeon might use the method presented in this study to determine the exact lesion location prior to surgery.

## Figures and Tables

**Figure 1 vetsci-11-00434-f001:**
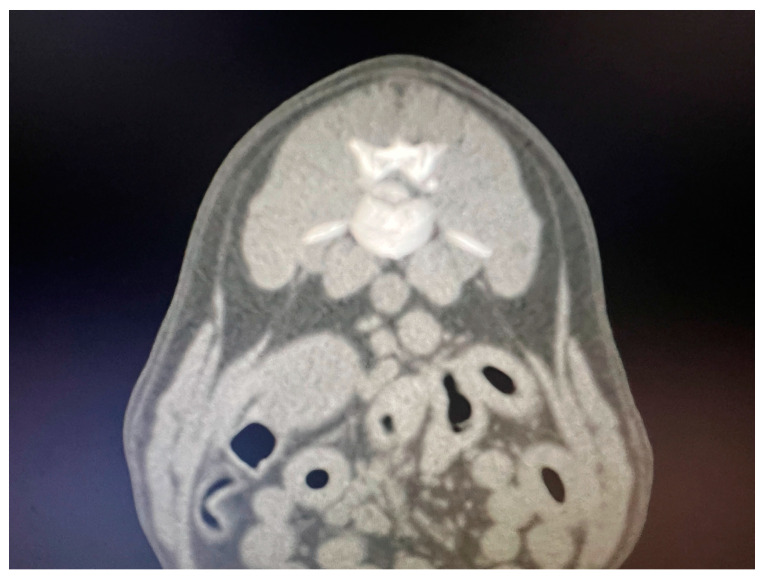
Transversal CT scan image showing a L1-L2 herniated disc.

**Table 1 vetsci-11-00434-t001:** Summaries of demographics by breed.

Variable	Category	French Bulldog (N = 32)	Dachshund (N = 18)	Shihtzu (N = 13)	Jack Russel (N = 11)	Other (N = 11)	*p*-Value
Age	1–6 years	25 (78%)	12 (67%)	7 (54%)	4 (36%)	7 (64%)	0.13
	6+ years	7 (22%)	6 (33%)	6 (46%)	7 (64%)	4 (36%)	
Gender	Male	18 (56%)	9 (50%)	8 (62%)	5 (45%)	7 (64%)	0.88
	Female	14 (44%)	9 (50%)	5 (38%)	6 (55%)	4 (36%)	
Grade	Grade 1	4 (13%)	2 (11%)	5 (38%)	1 (9%)	1 (9%)	0.29
	Grade 2	13 (41%)	8 (44%)	2 (15%)	3 (27%)	3 (27%)	
	Grade 3	9 (28%)	3 (17%)	3 (23%)	3 (27%)	0 (0%)	
	Grade 4	4 (13%)	2 (11%)	3 (23%)	3 (27%)	5 (45%)	
	Grade 5	2 (6%)	3 (17%)	0 (0%)	1 (9%)	2 (18%)	

Legend: grade 1—pain only; grade 2—nonambulatory paraparesis; grade 3—paraplegia with nociception; grade 4—paraplegia without nociception less than 48 h; grade 4—paraplegia without nociception more than 48 h. Summary statistics are number (percentage) or mean ± standard deviation.

**Table 2 vetsci-11-00434-t002:** Examination of learning effects for thoracolumbar data.

Outcome	Cases 1–15 (N = 15)	Cases 16–31 (N = 16)	Cases 32–47 (N = 15)	Cases 48–60 (N = 14)	*p*-Value
Correct 1st	4 (27%)	6 (38%)	7 (47%)	9 (64%)	0.04
Correct 2nd	7 (47%)	12 (75%)	12 (80%)	12 (86%)	0.02
Correct 3rd	12 (80%)	16 (100%)	15 (100%)	14 (100%)	0.02
Correct 4th	15 (100%)	16 (100%)	15 (100%)	14 (100%)	-
No of attempts	2.5 ± 1.1	1.9 ± 0.8	1.7 ± 0.8	1.5 ± 0.8	0.005

**Table 3 vetsci-11-00434-t003:** Examination of learning effects for lumbar data.

Outcome	Cases 1–8 (N = 8)	Cases 9–16 (N = 8)	Cases 17–25 (N = 9)	*p*-Value
Correct 1st	5 (63%)	6 (75%)	8 (89%)	0.20
Correct 2nd	7 (88%)	7 (88%)	9 (100%)	0.33
Correct 3rd	8 (100%)	8 (100%)	9 (100%)	-
Correct 4th	8 (100%)	8 (100%)	9 (100%)	-
No. of attempts	1.5 ± 0.8	1.4 ± 0.7	1.1 ± 0.3	0.22

Summary statistics are number (percentage) or mean ± standard deviation.

## Data Availability

The data presented in the study are included in the article. Further information can be obtained from the corresponding author.

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
