# Peer review of "Accurate Preoperative Localization of Thoracolumbar Disc Extrusion in Dogs: A Prospective Controlled Study"

_vetsci, 2024, doi:10.3390/vetsci11090434_

Round 1

Reviewer 1 Report

Comments and Suggestions for Authors

In connection with the manuscript submitted for review, “Accurate localization of thoracolumbar disc extrusion in dogs: a prospective controlled study,” I have the following comment:

 The topic is current and will become increasingly important over time, considering the role of pets in human life and their healthcare. The presented study lacks the necessary scientific context. The manuscript needs to be rewritten.

Introduction: The authors use humans as an example, so it would be beneficial to provide more information about achievements in this field, as this will likely be the trend for development in animals—dogs and cats.

 Materials and Methods: It is mentioned that CT and MRI were conducted; therefore, the results section should include figures with images from these studies (DOI: 10.7759/cureus.45628).

 Statistics should be based on the data presented in Table 1. The discussion does not correlate with the presented results, which need to be revised as mentioned above.

Discussion: Include data from publications concerning the issue in dogs. The results should be discussed (with new statistical data as noted above and based on the existing scientific literature). The current text in the “Discussion” can remain, but a new discussion should be added according to scientific requirements—discuss the results from the statistics, whether there is a correlation between sex, age, breed; discuss the potential for genetic predisposition, as is the case in humans (DOI: 10.7759/cureus.45628), with the appropriate literature sources, which are currently too few for such a relevant issue (https://scholar.google.com/scholar?q=thoracolumbar+disc+extrusion+in+dogs&hl=en&as_sdt=0,5).

 In this context, the manuscript will transform into a scientific article or a  case report with a review of a literature . The currently presented Brief Report does not provide the necessary scientific information, although it is well described for practical veterinary medicine purposes.

New statistics, extended introduction and new discussion are needed as I have indicated above.

Author Response

Thank you for your comments which offer us a chance to improve the quality of this Brief Report.

All the requested changes are highlighted in yellow in the manuscript.

In connection with the manuscript submitted for review, “Accurate localization of thoracolumbar disc extrusion in dogs: a prospective controlled study,” I have the following comment:

 The topic is current and will become increasingly important over time, considering the role of pets in human life and their healthcare. The presented study lacks the necessary scientific context. The manuscript needs to be rewritten.

Introduction: The authors use humans as an example, so it would be beneficial to provide more information about achievements in this field, as this will likely be the trend for development in animals—dogs and cats.

Response 1: Thank you for pointing this out. We agree with this comment. Therefore, we updated text in the manuscipt and it is highlighted in yellow. As we mentioned wrong level or wrong side surgery reports are rarely reported in the veterinary literature, likely due to biased reporting.  We expanded the conclusions of only the 2 reports which mentioned this topic.

Line 52-58: There are only two reports were this topic is mentioned [7-8]. One report [7] highlighted consequence of a bilateral mini-hemilaminectomy and intervertebral disc fenestration at T12-13 together with a bilateral pediculectomy of T13. Secondary subluxation and spinal cord compression five days postoperatively, required stabilization with bilateral articular screws and a dorsal spinal plate. The conclusion of the second report [8] was that operating on wrong side level does not always require a second revision surgery from the correct side to achieve spinal decompression.

 Materials and Methods: It is mentioned that CT and MRI were conducted; therefore, the results section should include figures with images from these studies (DOI: 10.7759/cureus.45628).

Response 2: Thank you for pointing this out. We agree with this comment. Therefore, we included a relevant CT image of one of the cases.

Line 76. Figure 1. Transversal CT scan image showing a L1-L2 herniated disc

 Statistics should be based on the data presented in Table 1. The discussion does not correlate with the presented results, which need to be revised as mentioned above.

Response 3: Thank you for pointing this out. We agree with this comment. Therefore, we included the following text, and we re-arranged the table 1, which now includes statistics.  At the discussion section we introduced correlations with the results.

Line 104-110: Analyses were performed to compare the characteristics of the different breeds. There were only a small number of dogs of some breeds, and thus to increase the numbers in each category those breeds with small numbers were combined together for analysis. Comparisons of age group and gender were compared between breeds using the Chi-square test. Grade was considered to be an ordinal variable, and to take account of the order of the categories, the analysis was performed using the Kruskal-Wallis test.

Line 77-80 Table 1: Summaries of demographics by breed

Variable

Category

French Bulldog

(N=32)

Dachshund

(N=18)

Shihtzu

(N=13)

Jack Russel

(N=11)

Other

(N=11)

P-value

Age

1 – 6 years

25 (78%)

12 (67%)

7 (54%)

4 (36%)

7 (64%)

0.13

6+ years

7 (22%)

6 (33%)

6 (46%)

7 (64%)

4 (36%)

Gender

Male

18 (56%)

9 (50%)

8 (62%)

5 (45%)

7 (64%)

0.88

Female

14 (44%)

9 (50%)

5 (38%)

6 (55%)

4 (36%)

Grade

Grade 1

4 (13%)

2 (11%)

5 (38%)

1 (9%)

1 (9%)

0.29

Grade 2

13 (41%)

8 (44%)

2 (15%)

3 (27%)

3 (27%)

Grade 3

9 (28%)

3 (17%)

3 (23%)

3 (27%)

0 (0%)

Grade 4

4 (13%)

2 (11%)

3 (23%)

3 (27%)

5 (45%)

Grade 5

2 (6%)

3 (17%)

0 (0%)

1 (9%)

2 (18%)

Summary statistics are: number (percentage) or mean ± standard deviation

Discussion: Include data from publications concerning the issue in dogs. The results should be discussed (with new statistical data as noted above and based on the existing scientific literature). The current text in the “Discussion” can remain, but a new discussion should be added according to scientific requirements—discuss the results from the statistics, whether there is a correlation between sex, age, breed; discuss the potential for genetic predisposition, as is the case in humans (DOI: 10.7759/cureus.45628), with the appropriate literature sources, which are currently too few for such a relevant issue (https://scholar.google.com/scholar?q=thoracolumbar+disc+extrusion+in+dogs&hl=en&as_sdt=0,5).

 In this context, the manuscript will transform into a scientific article or a  case report with a review of a literature . The currently presented Brief Report does not provide the necessary scientific information, although it is well described for practical veterinary medicine purposes.

New statistics, extended introduction and new discussion are needed as I have indicated above.

Response 4: Thank you for pointing this out. We agree with these comments. Therefore, we included the folowing text at discussion section.

The results are discussed now based on new Table 1 statistics and everything is highlighted in yellow. Line 150-154; line 167-169.

We introduced in discussion the recommended citations (DOI: 10.7759/cureus.45628; https://scholar.google.com/scholar?q=thoracolumbar+disc+extrusion+in+dogs&hl=en&as_sdt=0,5)

            Line 141-152. Wrong site surgery and wrong level surgery are reported in human neurosurgery, but currently veterinary neurosurgery has no specific reports related to incidence and management of such situations even if disc hernia is one of the most common neurologic problems encountered in veterinary clinical practice. [9]. Thus, the objective of this study was to assess the accuracy of disc localization prior to possible surgery for disk disease to ensure the surgeon begins their incision directly over the extruded disc. Of particular interest was whether surgeons would be able to preoperatively pinpoint the correct site for the surgery.

            Line150-154: The results of our study suggested there is no significant trend over time for any of the outcomes examined. There were some slight improvements over time, but none of     the results was close to statistical significance. The surgeons were able to have sequential multiple attempts until they got the site correct. Fewer errors in the assessment of neurolocalization occurred in the distal intervertebral spaces of the thoracic spine.

Line 167-169: Related to demographic characteristics of the dogs in the study, and comparisons between the different breeds, the analysis results indicated no significance between the different breeds for either age, gender or grade.

We want also to highlight the fact that we are aware of study limitations, thus we considered it as Brief Report and not case report or review of the literature. These are beyond our objectives.

Reviewer 2 Report

Comments and Suggestions for Authors

The article deals with a very interesting and innovative topic and forces an important reflection on the fundamental errors concerning the determination of the neurolocalisation of a lesion prior to neurosurgery. Such errors directly affect the success of the operation and the probability of recovery of the operated animal.   I read the article with great interest and strongly support the addressing of the above topic within the scope of clinical activities in veterinary medicine.   However, the article in my opinion needs a number of changes:

Firstly, the title does not fully reflect the subject matter of the article- in my opinion it should mention that this is a preoperative localisation assessment or that the article identifies the frequency of incorrect localisation assessment prior to surgery.  The materials and methods should also be expanded - there is no information regarding the period of the study (whether it is a dataset of several years, months, weeks).  Table 1 should be expanded to include information on the diagnostic modality (CT/MRI), and it would also be worth considering expanding the information to include the specific neurolocalisation (intervertebral space number). I assume that fewer errors in the assessment of neurolocalisation will occur in the distal intervertebral spaces of the thoracic spine.  There is also no information on the experience of the neurolocal markers, their specialisation and the number of people assessed in the experience. If the experience included one or two people then the result has no representative value.  The discussion should be extended to analyse the potential causes of the errors and compare them with solutions used in human medicine. 

Summary: The topic of this article has great clinical potential and raises a very important issue. However, the manuscript itself requires a significant number of revisions and, in my opinion, cannot be published in this form. 

Author Response

Thank you for your valuable comments which gives us the opportunity to improve this Brief Report. All changes are highlighted in yellow in manuscript.

The article deals with a very interesting and innovative topic and forces an important reflection on the fundamental errors concerning the determination of the neurolocalisation of a lesion prior to neurosurgery. Such errors directly affect the success of the operation and the probability of recovery of the operated animal.   I read the article with great interest and strongly support the addressing of the above topic within the scope of clinical activities in veterinary medicine.   However, the article in my opinion needs a number of changes:

Firstly, the title does not fully reflect the subject matter of the article- in my opinion it should mention that this is a preoperative localisation assessment or that the article identifies the frequency of incorrect localisation assessment prior to surgery.

Response 1: Thank you for pointing this out. We agree with these comments. Therefore, we change title with: Accurate preoperative localization of thoracolumbar disc extrusion in dogs: a prospective controlled study.

The materials and methods should also be expanded - there is no information regarding the period of the study (whether it is a dataset of several years, months, weeks).  Table 1 should be expanded to include information on the diagnostic modality (CT/MRI), and it would also be worth considering expanding the information to include the specific neurolocalisation (intervertebral space number). I assume that fewer errors in the assessment of neurolocalisation will occur in the distal intervertebral spaces of the thoracic spine.  There is also no information on the experience of the neurolocal markers, their specialisation and the number of people assessed in the experience. If the experience included one or two people then the result has no representative value. 

Response 2: Thank you for pointing this out. We agree with these comments. Therefore, we expanded materials and methods section by adding new requested data. As we had to change the table 1 according to the other reviewer recommendations, we included in the text information on the diagnostic modality (CT/MRI) and  specific neurolocalization (intervertebral space number). Information on the experience of the neurolocal markers, their specialisation and the number of people assessed in the experience were also added.

Line 66-71 Dogs with complete medical records, neurological examination and spinal cord compression diagnosed by CT (n=60) or MRI (n=25) (Figure 1) during year 2022 and 2023 were included in this study (Table 1). Specific neurolocalization (intervertebral space number) was represented by T10-T11 (n=5), T11-T12 (n=31), T12-T13 (n=23), T13-L1 (n=16), L1-L2 (n=5), L2-L3 (n=3), L3-L4 (n=2). A board certified surgeon (WMC) and two residency-trained surgeons (CO and CY) performed neurological localization attempts and surgeries.

Line 153-154: Fewer errors in the assessment of neurolocalisation occured in the distal intervertebral spaces of the thoracic spine.

The discussion should be extended to analyse the potential causes of the errors and compare them with solutions used in human medicine. 

Thank you for pointing this out. We agree with these comment. Therefore, added:

Response 3: In human medicine there are few recommendations to prevent WLS: such communication with the patient, marking of the intended site, and the use of intraoperative radiograph. [4]. This study showed that the technique for marking the intended site with novel solution is a viable source of prevent errors. Intraoperative radiographs is another potential solution for veterinary patients and future studies need to validate it.

Summary: The topic of this article has great clinical potential and raises a very important issue. However, the manuscript itself requires a significant number of revisions and, in my opinion, cannot be published in this form. 

We want also to highlight the fact that we are aware of study limitations, thus we considered it as Brief Report. Adding new data/discussion is limited by the nature of publication and transforming it in Research Paper is not our intention.

Round 2

Reviewer 1 Report

Comments and Suggestions for Authors

The authors have complied with the reviewers' recommendations and the authors have improved the quality of the submitted manuscript.

Reviewer 2 Report

Comments and Suggestions for Authors

The manuscript has been revised correctly in accordance with the suggestions made in the previous review. In my opinion, it can be published in this form.